# The Potassium-Induced Decomposition Pathway of HCOOH on Rh(111)

**Imre Kovács** [1,]*[ ], **János Kiss** [2] **and Zoltán Kónya** [2,3]

[1] Institute of Technology, University of Dunaújváros, Táncsics M. u. 1/A, 2401 Dunaújváros, Hungary
[2] MTA-SZTE Reaction Kinetics and Surface Chemistry Research Group, University of Szeged, Rerrich Béla Square 1, 6720 Szeged, Hungary; jkiss@chem.u-szeged.hu (J.K.); konya@chem.u-szeged.hu (Z.K.)
[3] Department of Applied and Environmental Chemistry, University of Szeged, Interdisciplinary Excellence Centre, Rerrich Béla Squer 1, H-6720 Szeged, Hungary
[*] Correspondence: kovacsimidr@gmail.com

**Abstract:** Formic acid (FA) can be considered both a CO and a $H_2$ carrier via selective dehydration and dehydrogenation pathways, respectively. The two processes can be influenced by the modification of the active components of the catalysts used. In the present study the adsorption of FA and the decomposition of the formed formate intermediate were investigated on potassium promoted Rh(111) surfaces. The preadsorbed potassium markedly increased the uptake of FA at 300 K, and influenced the decomposition of formate depending on the potassium coverage. The work function ($\Delta\phi$) is increased by the adsorption of FA on K/Rh(111) at 300 K suggesting a large negative charge on the chemisorbed molecule, which could be probably due to the enhanced back-donation of electrons from the K-promoted Rh into an empty $\pi$ orbital of HCOOH. The binding energy of the formate species is therefore increased resulting in a greater concentration of irreversibly adsorbed formate species. Decomposition of the formate species led to the formation of $H_2$, $CO_2$, $H_2O$, and CO, which desorbed at significantly higher temperatures from the K-promoted surface than from the K-free one as it was proven by thermal desorption studies. Transformation of surface formate to carbonate (evidenced by UPS) and its decomposition and desorption is responsible for the high temperature CO and $CO_2$ formation.

**Keywords:** formic acid decomposition; formate intermediate; potassium adatom; Rh surfaces

## 1. Introduction

Formic acid (FA), HCOOH, is an important chemical for the renewable energy system. FA can be decomposed to CO and $H_2O$ by dehydration or to $H_2$ and $CO_2$ by dehydrogenation. A third way gives green syngas by competitive decomposition [1–3]:

$$HCOOH = CO + H_2O \tag{1}$$

$$HCOOH = CO_2 + H_2 \tag{2}$$

$$HCOOH = CO + H_2 + O \tag{3}$$

The selective decomposition path of FA offers promising channels either for high purity CO, which is widely used in semiconductor industry [4] or for $H_2$ in high purity, which can be applied for fuel cell vehicles; FA is a good candidate as a $H_2$ storage compound [5–8]. The importance of FA for fuel cell application can be demonstrated nowadays by an increased number of patents. In addition, the FA decomposition can provide syngas with different $CO/H_2$ ratio for chemical synthesis

or applications [1]. FA can be synthetized by either the catalytic $CO_2$ hydrogenation [9,10] or the oxidation of biomass [11,12]. It was found that alkali cation ($Na^+$) near to Rh particles promote the conversion of $CO_2$ towards HCOOH production via formation of formate. The formate species and the hydride rhodium complexes are considered reaction intermediates in formic acid formation [10].

Formic acid, HCOOH, is a very useful precursor molecule in producing formate surface intermediate because it can be dehydrogenated on metal and oxide surfaces. The knowledge of the surface chemistry of adsorbed formate as a reaction intermediate is of great assistance in the elaboration of the mechanism of several important catalytic reactions which contain $C_1$ species [13] such as water -gas shift reaction [14,15], methanol synthesis [16–20] and methanation of CO [21,22] and $CO_2$ [23–34]. The formate species can be detected on supported metal catalysts during these reactions mainly with IR vibrational techniques. The IR spectra obtained after the adsorption of HCOOH were the same on supported metal (Rh) and on the support alone. It was suggested that formate ion formed in the above mentioned reactions, on metal sites, then it is re-located on the support, or the support basically influence the stability of the formate species on metals. Therefore, it is desirable to investigate its formation and stability on clean metal surfaces under UHV conditions without the disturbing effect of the support. For this purpose, the surface study of the dissociative adsorption of FA is an excellent topic.

It turns out from several studies that not only the support, but also the different additives have a marked influence on the decomposition of FA and the characteristics of the surface formate on different metal surfaces [35–42]. On rhodium surfaces, the attention is focused on the effect of electronegative (preadsorbed oxygen) and electropositive (potassium) additives. The interaction of formic acid with clean and oxygen-covered Rh(111) surfaces has been investigated by electron energy loss (in the electronic range), thermal desorption and photoelectron spectroscopy [43]. The formate species on clean Rh(111) was stable up to 200 K, but decomposed completely at 200–250 K. The major products were: $H_2$ and $CO_2$, but $H_2O$ and CO were also formed. Preadsorbed oxygen exerted a readily observable influence on the interaction of HCOOH with the Rh(l11) surface. It increased the extent of dissociation of FA and extended the region of stability of surface formate by at least 80–100 K. This was demonstrated by the higher stability of photoemission peaks of formate and by the simultaneous production of $CO_2$ and $H_2O$ with $T_p$ = 377–385 K at saturation oxygen coverage. CO production was not observed. The effect of boron contamination on the Rh foil is different, in this case the boron segregating from the bulk interacts with the products ($H_2O$, $CO_2$), therefore it increases the amount of CO and $H_2$ [44–46].

The effect of an electropositive potassium additive on the decomposition of HCOOH on Rh(111) was studied previously [47,48]. The photoelectron [UPS] and thermal desorption spectroscopic study was restricted only to the low temperature region (adsorption temperature was 100 K). The preadsorbed potassium promoted the dissociation of FA and increased the surface concentration of the most stable formate anion. In the light of the obtained results, it is desirable to investigate the interaction between the potassium and the formate at high temperatures, and also at different potassium and FA coverages on Rh(111). In the present study ultraviolet photoelectron spectroscopy (UPS), work function ($\Delta\phi$) and thermal desorption spectroscopy (TDS) studies were carried out.

## 2. Results

### 2.1. Thermal Desorption Measurements

In the first series of measurements, we investigated the effects of the potassium coverage on the desorption of HCOOH, and on the formation of the decomposition products. Molecular FA desorption was not observed after adsorption at 300 K on clean Rh(111). CO formation was detected with $T_p$ = 489 K (Figure 1A). $H_2$ appeared on TPD spectra with a weak, broad peak between 320–350 K. $CO_2$ and $H_2O$ as decomposition products were not observed after adsorption at 300 K. When these products are formed, they leave the surface immediately the clean surface. In our previous work,

where the adsorption temperature was 100 K, all $CO_2$ and $H_2O$ desorbed below 300 K. The transient surface intermediate was the formate species [43].

The effect of potassium coverage ($\Theta_K$) on the adsorption of FA at 300 K was investigated between ~0.1 and 0.36 coverages. At $\Theta_K$ = 0.36 the potassium reaches the saturation coverage (1 ML) [49]. The presence of potassium, with different coverages, altered the bonding mode of FA and the formation temperature of the products. Both the work function changes ($\Delta\phi$) and the core level peak shift (XPS) support the notation of a considerable charge transfer from K to Rh at low coverages, and a gradual neutralization at saturation [49–54]. At $\Theta_K$ < 0.15 the potassium is positively charged ($K^+$), while at $\Theta_K$ = 0.36 the potassium exhibits mainly metallic character.

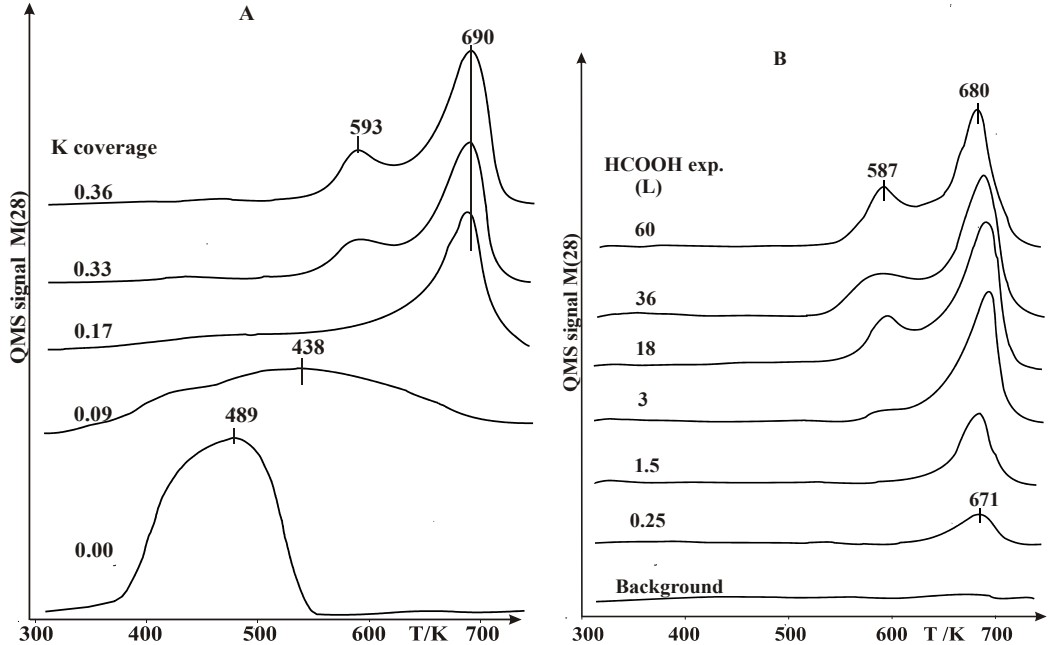

**Figure 1.** (**A**) Thermal desorption spectra of CO from different potassium covered Rh(111) FA exposure was 12 L. (**B**) Thermal desorption spectra of CO as a function of HCOOH exposure at monolayer K coverage ($\Theta_K$ = 0.3).

Figure 1A shows the TD spectra of CO after adsorption of FA at different potassium coverages. At $\Theta_K$ = 0.09, where the potassium exhibits ionic character, a very broad feature was detected. At a higher K concentration the potassium acquires a metallic-like character, and the CO desorption temperature shifted to 593 and 690 K. Detailed CO desorption spectra are displayed at $\Theta_K$ = 0.36 as a function of FA exposures on Figure 1B. Very similar desorption features were detected after CO was adsorbed on potassium-dosed Rh(111) [55–57]. The results indicate both strengthening of the M-C bond and the weakening of the C-O bond in the presence of potassium, due presumably to an increased electron occupancy of the $2\pi^*$-orbital of CO. Broadening and asymmetry of the vibrational peaks suggest that the proximity of the CO molecules to the potassium adatoms influence the chemisorption behavior, although nonlocal interactions are also indicated. Hydrogen desorbs independently of the potassium coverage at $T_p$ = 345 K (Figure 2A), at high exposures a tailing can be seen at the high temperature side (~450 K). As follows from the results plotted in Figure 2B, the desorption of $H_2O$ occurred in new high temperature states ($T_p$ = 440 and 500–600 K) from K covered Rh(111) These states could be associated with the potassium stabilized water desorption and adsorbed OH recombination especially at around monolayer potassium coverage. Strong interaction between K and water including dissociation was established previously, and a strong thermodynamic driving force to KOH formation ($\Delta H_{f(KOH)}$ = −424.7 kJ/mol) was calculated [57]. The amount of $H_2O$ formed during desorption significantly increased with potassium coverages at the same FA exposure.

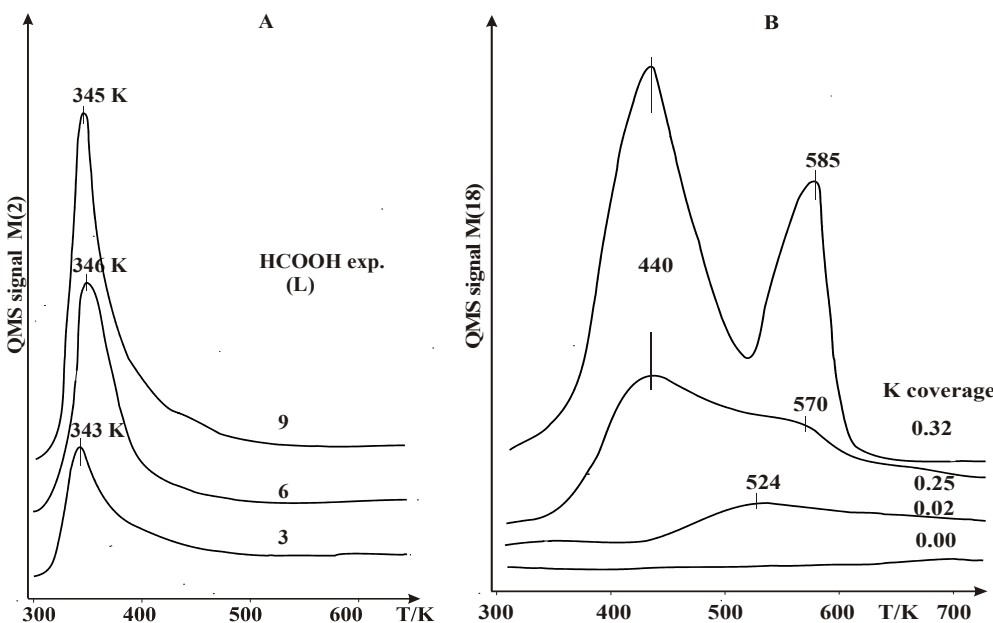

**Figure 2.** (**A**) Thermal desorption spectra of $H_2$ from monolayer potassium coverage, $(\Theta_K = 0.3)$.
(**B**) Thermal desorption spectra of $H_2O$ from different potassium covered Rh(111).

The $CO_2$ TD spectra, after adsorption of FA at 300 K, are more complex, due to the desorption and different steps of decomposition of surface complexes. Significant differences can be seen at different potassium coverages. TDS for $CO_2$ at $\Theta_K = 0.1$ are displayed in Figure 3A.

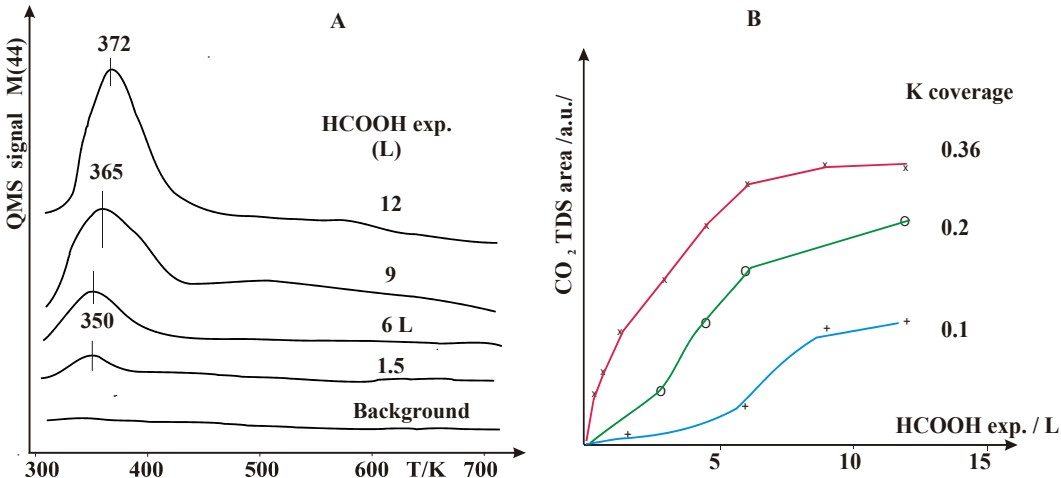

**Figure 3.** (**A**) Thermal desorption spectra of $CO_2$ from low potassium coverage, $(\Theta_K = 0.1)$. (**B**) The amount of desorbed $CO_2$ as a function of potassium coverage after different HCOOH exposures at 300 K.

One dominant peak was detected; its peak temperature shifted from 350 K to 372 K with increasing HCOOH exposure. Very probably, the potassium stabilized the formate species via an electronic interaction [37–40], and then the stabilized formate decomposes at this temperature giving $CO_2$ and $H_2$. As the potassium coverage was increased the stabilization effect to formate also increased, very probably surface compounds are formed. The amount of $CO_2$ formed during desorption significantly increased with the potassium coverage. The areas of the desorption peaks are plotted as a function of FA exposures at different potassium coverages in Figure 3B. The effect of potassium was exhibited in the higher desorption temperatures of decomposition products. The TD spectra obtained at $\Theta_K = 0.2$ shows an additional high temperature desorption peak; the first two peaks developed at 672 and

705 K. When the FA exposure was increased to 6 L, a weak peak appeared at $T_p$ = 466 K, and an intense one developed at 581 K (Figure 4A). The TD spectra of $CO_2$ as a function of FA exposures at monolayer potassium coverage ($\Theta_K$ = 0.3), at which the potassium has a metallic character, are displayed in Figure 4B. At low FA exposures, 0.3–0.6 L, two high temperature peaks with $T_p$ = 654 K and $T_p$ = 675 K were developed. From 3 L exposure, the intensity of these peaks increased, and a third peak was developed at $T_p$ = 589 K. It is important to mention again that CO desorption happened also at $T_p$ = 587 K and $T_p$ = 680 K. These coinciding temperatures in the CO and $CO_2$ desorption strongly suggest the same source of products. It should be emphasized that potassium stabilization also occurs in the co-adsorbed layer, in this potassium desorption was detected at $T_p$ = 587 K and $T_p$ = 680 K [49]. Similar coincidence temperatures were observed after $CO_2$ adsorption from K/Rh [49,58] and K/Pd surfaces [59].

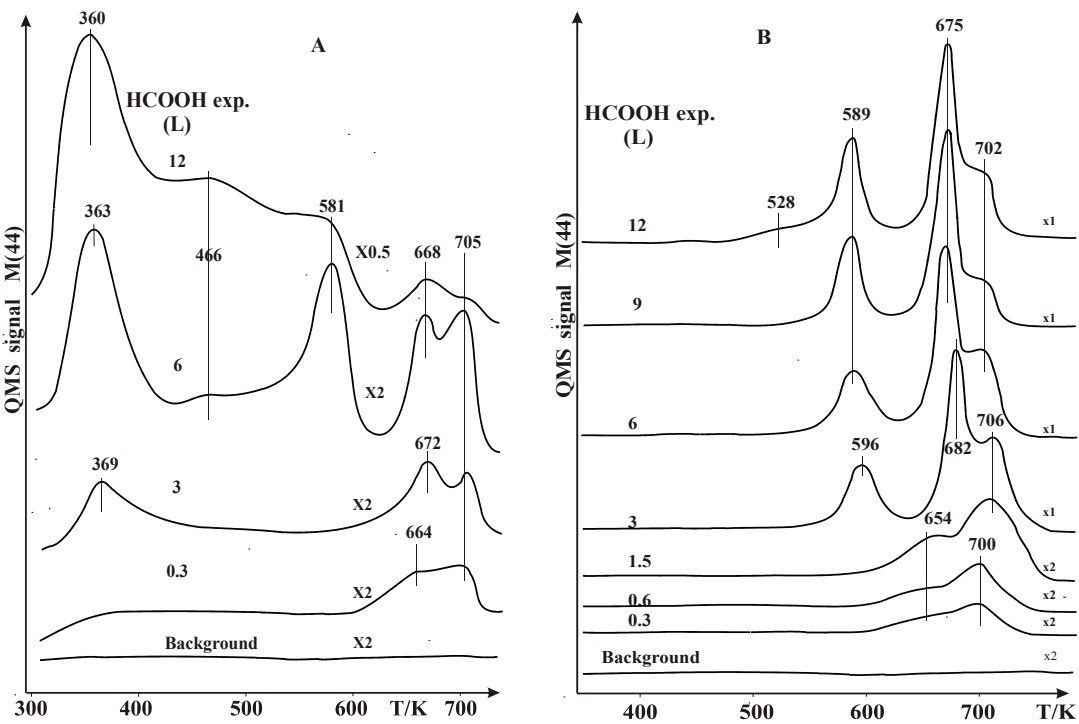

**Figure 4.** (**A**) Thermal desorption spectra of $CO_2$ at $\Theta_K$ = 0.2. (**B**) $CO_2$ desorption spectra at monolayer K coverage, ($\Theta_K$ = 0.3) as a function of HCOOH exposure.

### 2.2. Work Function ($\Delta\phi$) and UPS Measurements

In order to help understand the mechanism of the surface reactions in the co-adsorbed layer, work function and photo electron spectroscopic measurements (UPS) were carried out. The work function changes observed following potassium deposition on a clean Rh(111) surface were reported previously [49]. The work function of Rh decreased linearly with K exposure up to $\Theta_K$ ~ 0.15, ($\Delta\phi$ = −3.5 eV). Further K deposition led to a slight increase (0.5 eV) in ($\Delta\phi$). The large linear decrease in the work function at low potassium coverages indicates the formation of a species with high dipole moment; the formation of an ionic K. Above $\Theta_K$ = 0.15, a strong dipole-dipole depolarization starts to compensate the effect of the increasing K concentration, (formation of metallic potassium). Independently of the potassium coverage the adsorption of FA at 300 K resulted in work function increases (Figure 5A). The extent of increase is proportional with the K coverage. Heating of the coadsorbed layer resulted in a complex picture (Figure 5B). A decrease in the work function corresponds to the desorption and decomposition of adsorbed HCOOH species. Above 600 K the work function started to increase slowly. The original value for the clean Rh surface was attained only above 900–1000 K.

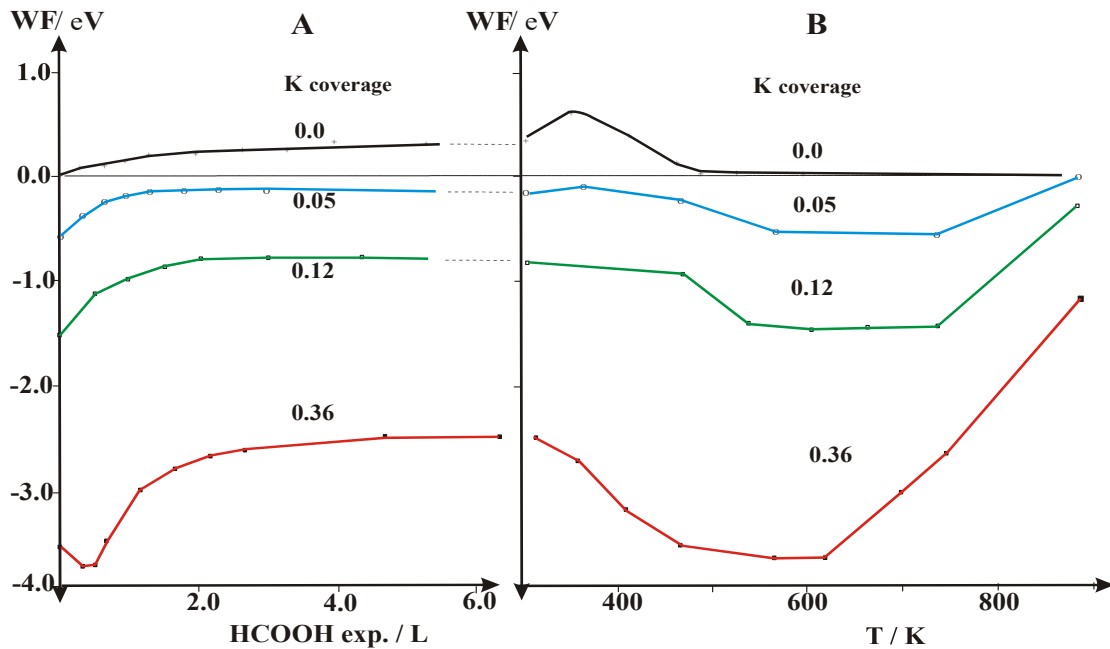

**Figure 5.** (**A**) Change of the work function, ($\Delta\phi$), at different potassium coverage with FA exposure. (**B**) Change the work function during heating the adsorbed layer.

The photoemission spectra of adsorbed HCOOH were taken on a clean and K-dosed Rh(111) surface at $\Theta_K = 0.3$. The observed photoemissions obtained after HCOOH adsorption are collected in Table 1. Figure 6 shows the HeII UP spectra obtained on clean surface at different FA exposures. FA adsorption at 300 K resulted in two photoemission peaks at 8.2 and 11.5 eV, which correspond to the $5\sigma/1\pi$ and $4\delta$ orbitals of CO, respectively. These orbitals can be detected also after CO adsorption on a clean surface [55,56]. Their intensities remained constant up to 395 K, then they decreased and disappeared around 510 K. Photoemissions attributable to formate at 5.3, 8.6, 10.2 and 14.2 eV [43,60,61] were not observed. This result supports the hypothesis that the formate is not a stable intermediate above 300 K on a clean surface.

**Table 1.** Binding energies (in eV) of adsorbates observed following HCOOH on Rh(111) and K/Rh(111).

|  | $\theta_K$ | UPS | References |
|---|---|---|---|
| HCOOH only at 100K | 0 | 6.2 8.9 10.5 11.9 | [43] |
|  | 0.1 | 6.2 9.1 10.5 12.0 | [48] |
|  | 0.33 | 6.2 9.1 10.5 12.0 | [48] |
| $HCOO^-$ | 0 | 5.3 8.9 10.2 13.2 | [43,60,61] |
|  | 0.1 | 5.2 8.9 10.2 14.2 | [48] |
|  | 0.33 | 5.2 8.9 10.3 14.0 | [48], this work |
| $CO_3^{2-}$ | 0.33 | 8.4 10.2 | [48,62] this work |
| CO | 0 | 8.0 10.8 | [55,56] |
|  | 0.1 | 8.0 10.9 | [48] |
|  | 0.33 | 9.0 11.5 | [48,51], this work |
| O | 0 | ~6 | [57] |
|  | 0.33 | 5.2 | [56,57], this work |

UPS after FA adsorption at 300 K and subsequent heating are presented in Figure 7. When the FA was introduced to the potassium-covered Rh(111) at $\Theta_K = 0.3$, which corresponds to monolayer coverage, the ultraviolet spectrum was complex at 300 K (Figure 7). On this potassium-covered surface peaks were found at 5.4, 8.9, 10.3, 14.2 eV, which correspond, to the $6a_1$, $4b_2$, $5a_1$ and $4a_1$ orbitals of formate, respectively [43,60,61]. Above 420 K this adsorption form cannot be detected above 507 K. The strong peaks are due to adsorbed CO found up to 705 K at 8.1 and 11.5 eV, which are characteristic

of adsorbed CO on potassium-covered Rh(111) [48,51,56]. From 507 K shoulders are visible at ~8.4 and 10.3 eV. These emissions can be attributed to surface carbonate and are detectable up to 640 K. The weak feature could be due either to the low concentration of this surface compound or to the strong overlapping of the CO 5σ/1π orbital with the unresolved combination of 3e′/1a″ molecular orbitals of the $CO_3$ species at 8.4 eV. The observed photoemission peaks conform well to those obtained for other carbonate species (3e′.1a″ unresolved and 4a′ orbitals) including $K_2CO_3$ [48,62]. From 640 K an additional emission showed up at ~5.0 eV which could be attributed to adsorbed oxygen bonded to potassium ($K_2O$) [56,57].

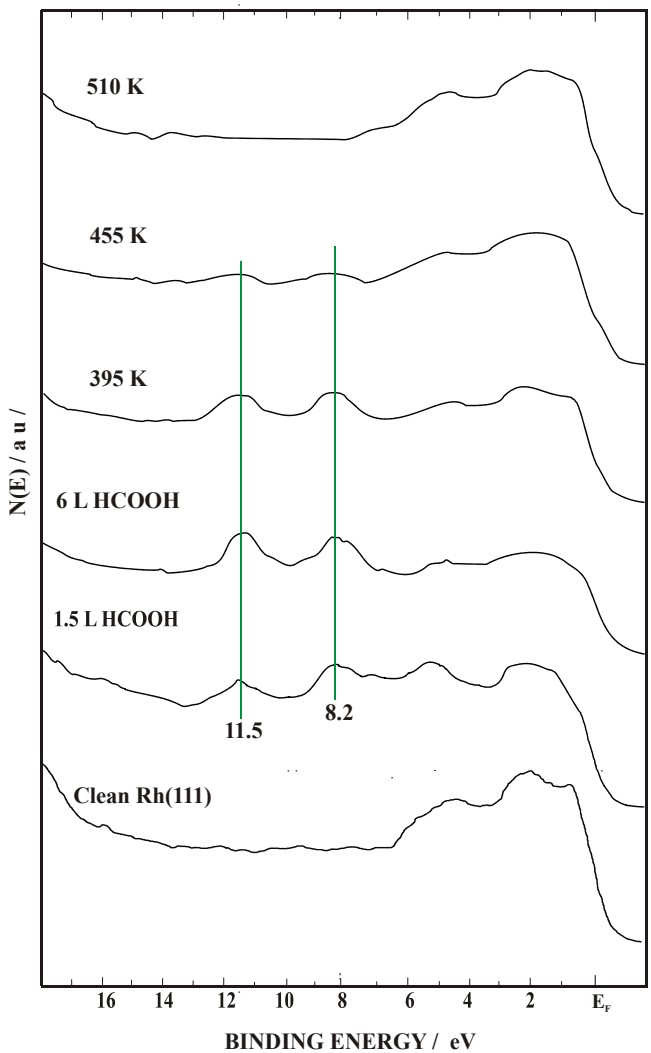

**Figure 6.** UP spectra after adsorption of FA on clean Rh(111) at 300 K and subsequent heat treatment. FA exposure was 12 L.

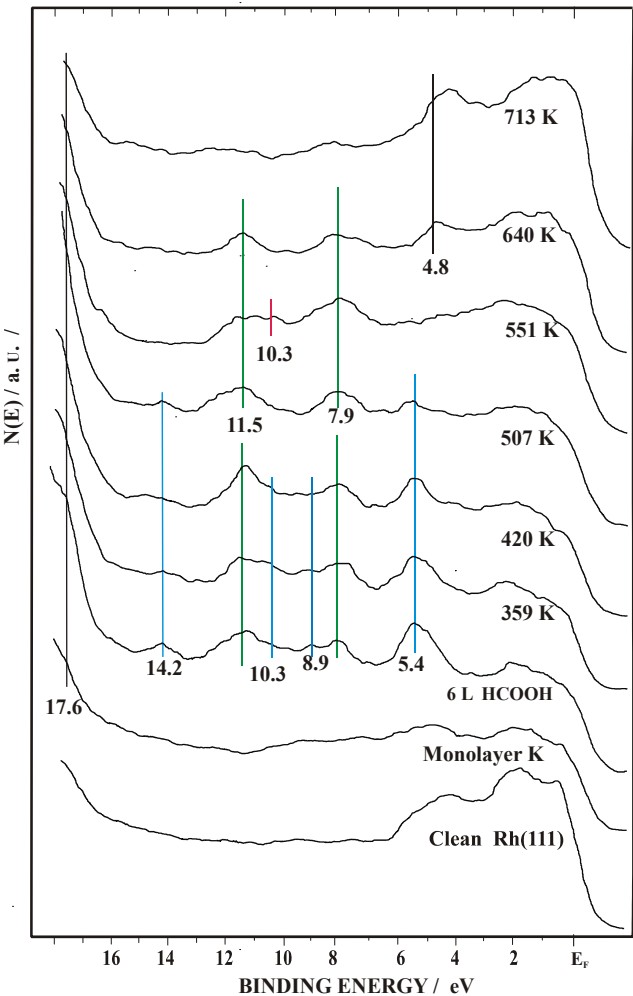

**Figure 7.** UP spectra after adsorption of FA on potassium covered Rh(111) at monolayer K coverage, $\Theta_K$ = 0.3 at 300 K and subsequent heat treatment. FA exposure was 12 L.

## 3. Discussion of Surface Reaction Mechanism

The study of the decomposition of HCOOH (FA) and the surface chemistry of formate (HCOO) is receiving increasing attention nowadays. As we demonstrated in the Introduction, the selective decomposition path of FA offers promising channels to access high purity $H_2$ production and FA is a good candidate as a $H_2$ storage compound [2–4]. The classic Sachtler–Fahrenfort volcano curve for HCOOH decomposition by metal powder catalysts is revisited with application of a modern catalysis approach. The Rh is positioned around the top of the volcano curve [63]. From a catalytic point of view, FA is a very useful precursor molecule in producing formate surface intermediates formed in several catalytic reactions. Formate could be an important intermediate in $CO_2$ methanation [31–34,64]. Formate may also be hydrogenated to methoxy species, which produces methanol [16–20]. Both formate [37,48,65] and methoxide [66,67] can be stabilized by alkali promotors. Rh is one of most investigated metals for $CO_2$ hydrogenation reactions [31,32]. In this, reaction the formate intermediate may decompose to CO, or on the other hand, the formate may be further hydrogenated producing $CH_4$. Moreover, the production of CO and the methane on supported Rh catalysts depend upon the temperature, pressure, and presence or absence of promotor. Addition of Ba and K to the $Al_2O_3$ support results in significant differences in the catalytic behavior [31,68]. $CH_4$ was preferentially formed on Ba-containing and pure $Rh/Al_2O_3$ and only CO was observed with K-containing catalysts.

All these findings motivated us to focus experimentally on the surface chemistry of HCOOH/formate on Rh without support. The ongoing interest in the surface chemistry of FA

has also attracted recently extensive density functional theory (DFT) calculations for HCOOH decomposition on slabs of modified metal surfaces with different facets [69]. Different additives (B, O, alkali metals and $NH_4$) significantly alter the decomposition pathway of FA and formate on metal surfaces [35,37–40,42,43,48,70]. Electropositive adatoms (Cs, K, $NH_4$) on metals enhance the extent of dissociation of FA [37–40,48,70]. In the present study, the adsorption and decomposition of FA were studied on potassium-promoted Rh(111). The adsorption temperature was 300 K. We demonstrated that this adsorption temperature caused a higher dissociation and it altered the product (CO, $CO_2$, $H_2$ and $H_2O$) distribution. The promotor effect was significantly higher at 300 K adsorption then at lower temperature (100 K). The quantity of the products formed during desorption proportionally increased with potassium coverages, the product distribution was also altered at different potassium concentrations.

The work function ($\Delta\phi$) increases during the adsorption of FA on K/Rh(111) (Figure 5). This suggests a large negative charge on the chemisorbed molecule, which is probably due to the enhanced back-donation of electrons from the potassium-promoted Rh into an empty $\pi$ orbital of HCOOH. We may assume that this enhanced back-donation occurs directly between the formate and the K/Rh surface. The binding energy of the formate species is therefore increased. This would result in a greater concentration of irreversibly adsorbed formate species, and this found in TD studies. At low potassium coverage, where the potassium is of fully ionic character, we suggest to the following steps of formate decomposition from the observed TD spectra:

$$HCOOH_{(a)} + K_{(a)}^+ = \{HCOO^- + K_{(a)}^+\} + H^+_{(a)} \tag{4}$$

$$\{HCOO^- + K_{(a)}^+\} + H^+_{(a)} = H_{2(g)} + CO_2 + K^+_{(a)} \tag{5}$$

$$\{HCOO^- + K_{(a)}^+\} + H^+_{(a)} = CO_g + H_2O_{(g)} + K^+_{(a)} \tag{6}$$

At higher potassium coverages the interpretation of TD spectra is more complex due to the various desorption states. It cannot be excluded that potassium at certain coverages may stabilize the transiently produced $CO_2$ in dimer form. This dimer may disproportionate to carbonate and CO or releases $CO_2$ with $T_p$ = 466 K. It is clearly visible at $\Theta_K$ = 0.2 (Figure 4A). At monolayer K coverages the interaction of formate and potassium could be stronger, and potassium formate is formed. The first step of decomposition of formate the oxalate formation route in which $H_2$ release below 450 K. In our UP spectra, formate can be detected up to 507 K (Figure 7). This compound decomposes above 550 K giving CO and $CO_2$ desorption with Tp = 589 K. At the same time $H_2O$ evolution was also detected. Potassium formate is stable compound [71], its decomposition is described by two parallel occurring reactions via oxalate and carbonate formation according to the literature [71]. It is important to note that carbonate appeared in our UP spectra around 551 K at 8.4 and 10.3 eV (Figure 7). Accepting these observations, we conclude that the potassium formate decompose similar way on the potassium covered Rh surfaces, too:

$$2HCOOK_{(a)} = K_2C_2O_{4(a)} + H_{2(g)} \tag{7}$$

$$2HCOOK_{(a)} = K_2CO_{3(a)} + H_2O_{(g)} \tag{8}$$

$$K_2C_2O_{4(a)} = K_2CO_{3(a)} + CO_{(g)} \tag{9}$$

$$K_2CO_{3(a)} = K_2O + CO_{2(g)} \tag{10}$$

CO liberates from oxalate-carbonate transformation, the $CO_2$ releases from carbonate decomposition. The CO and $CO_2$ show up in gas phase at somewhat higher temperature then steps (6) and (7) occur, because the adsorbed potassium may further stabilize them. The highest temperature desorption peak of $CO_2$ ($T_p$ = 702 K) may also corresponds to K-$CO_2$ decomposition formed in direct interaction of two species. This feature was found after $CO_2$ adsorption on potassium-covered Rh(111) and Pd(100) surfaces [42,52]. The appearance of the photoemission peak at ~5 eV around

640 K may be attributed to the formation of $K_2O$. The schematically illustration of decomposition of potassium stabilized formate intermediate is displayed in Figure 8.

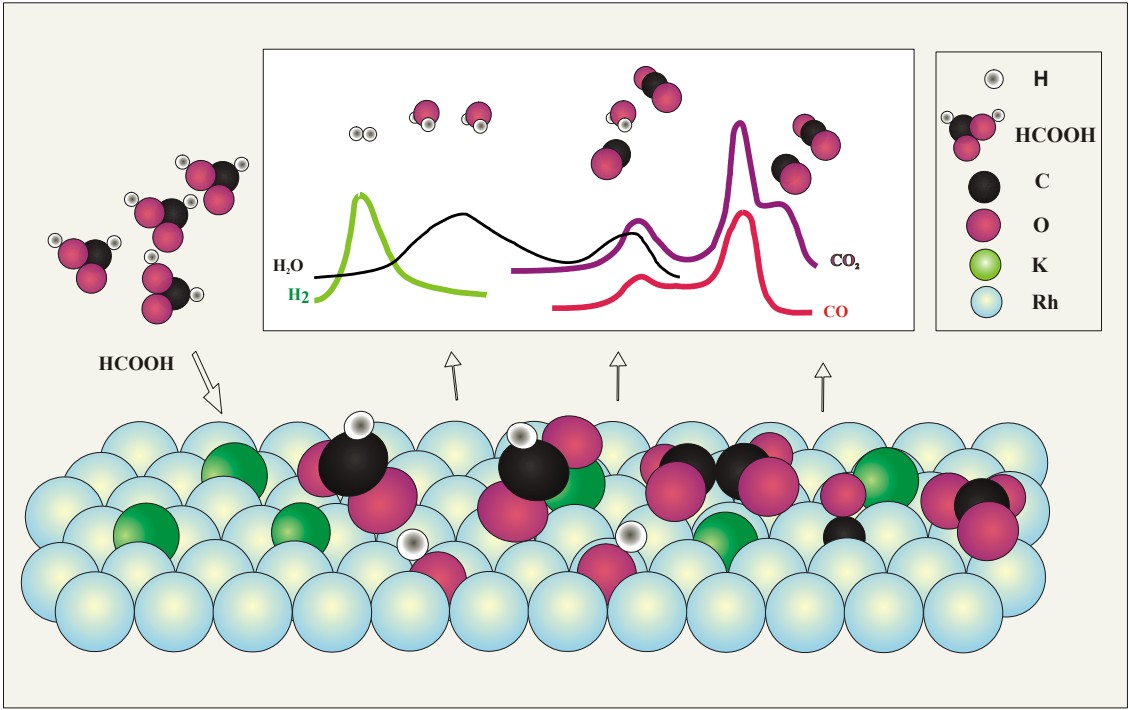

**Figure 8.** Schematic illustration of formate decomposition on potassium promoted Rh(111) surfaces.

## 4. Experimental

Experiments were performed in a stainless steel UHV chamber equipped with an electron source for Auger electron spectroscopy (AES), a photon source (He I, He II) for UPS which was pumped differentially, and a quadrupole mass spectrometer for thermal desorption spectroscopy (TDS). An electrostatic hemispherical energy analyzer (LRS 10, Leybold-Hereaus, Cologne, Germany) detected auger and photoelectrons. Changes in work function (($\Delta\phi$) were obtained from the He I UPS spectra. TDS were taken in "line of sight" with a heating rate of 10 Ks$^{-1}$. The Rh crystal was cut from a single crystal. It was a product of Materials Research Corporation (Orangeburg, Route 303, NY, USA); the purity was 99.99%. The surface orientation was determined by LEED before the present measurements in a separate chamber. The sample was heated resistively, and its temperature was measured by a chromel-alumel thermocouple spot-welded to the edge of the crystal. The cleaning procedure contains argon ion bombardment (600 eV, $1 \times 10^{-6}$ mbar Ar, 3 μA for 10–30 min), and annealing at 1270 K for some minutes. Contaminations including boron impurity were not detected by AES. A commercial SAES Getter alkali metal source (Lainate, Milan, Italy) was used to deposit K. The K coverage was determined by means of AES and TDS. The determination of K coverage is described in our previous works [48,51].

## 5. Conclusions

Formic acid (FA) decomposes on clean Rh(111) at room temperature leaving only CO on the surface. Potassium adatom basically altered the reaction pathway. Potassium stabilizes the transiently formed formate species. At low potassium coverage ($\Theta_K < 0.15$), where the K is fully ionized, a two dimensional {$HCOO^- + K_{(a)}^+$} surface complex is formed, which decomposes to $CO_2$, CO, $H_2$ and water. At monolayer potassium coverage ($\Theta_K \sim 0.33$), the formate stabilized by potassium transforms into surface compounds (oxalate, carbonate) in consecutive reactions, while $H_2$ and $H_2O$ are released. Finally CO and $CO_2$ are liberated from the surface. Potassium was also stabilized by coadsorbates.

**Author Contributions:** All authors contributed to write the paper. All authors have read and agreed to the published version of the manuscript.

**Funding:** This research received no external funding.

**Acknowledgments:** The authors wish to thank Albert Oszkó for the careful revision of the manuscript. Financial support of this work by the National Research Development and Innovation Office through grant NKFIH OTKA K120115 (Zoltán Kónya).

**Conflicts of Interest:** The authors declare no conflict of interest.

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
