# Peer review of "The Potassium-Induced Decomposition Pathway of HCOOH on Rh(111)"

_catalysts, doi:10.3390/catal10060675_

Round 1
Reviewer 1 Report
Quality of this manuscript is consistent with the journal. My only criticism is poor quality of graphical material.
Chemical equations should be aligned on the left side of each other. The system of equations should be centered (page 1 and 10 of the manuscript).
Some figures (fig. 1 – 7) seems just a scan-copies. I strongly recommend to prepare figures in colour format by using modern software. Otherwise, the manuscript will be looking old-fashioned.
Fig 9 does not have a caption. The quality of this figure should be improved.
The last part of the manuscript looks a bit unsuccessful. The last word should not be Fig. 8. I recommend to add at least brief conclusion.
Scientific content of the manuscript is interesting. The paper can be accepted to publication in Catalysts after minor technical revision.
Author Response
Answer to Reviewer #1
First, we thank the review of our manuscript and your suggestions. We accepted them and we made corrections accordingly. Technical revision was done. We improved the quality of figures.
- 1-7 are improved.
- We supplied the Fig. 8 with a caption.
- We added a brief conclusion at the end.
We very hope that our revised version is suitable for publication.

Reviewer 2 Report
The work entitled „Potassium-induced pathway of decomposition of HCOOH on Rh(111)” by Imre Kovács, János Kiss and Zoltán Kónya reveals an interesting problem of surface phenomena on Rh(111) related to the adsorption and transformation of formic acid.
Generally, the work is well organized, the data are reliable and the conclusions are well supported by the results of the studies. The topic is important and attractive. However, the main weakness of this work is the lack of reference to similar studies not only in adsorption discussion (what is present) but also in the overall efficiency and selectivity of the process. There is no benchmark based on common catalysts. This deficiency unfortunately makes it difficult to assess the value of the results.
There are also some too obvious statements, which move the discussion on a basic level, for example: “It is known from inorganic chemistry that potassium formate is stable compound [56].”
Some inconsistency could be found in funding information (“The research received no external funding.”), while in the next paragraph one of the Authors acknowledges “National Research Development and Innovation Office through grant 297 NKFIH OTKA K120115”
Apart from that, the work has numerous spelling errors, for example:
- 1 “which can applied be”
- 7 “Above 420 K this adsorption form can not be detected”
- 9 “production and FA isa good”
Graphics is of low quality, which should be definitely improved.
My recommendation is a major revision.
Author Response
Answer to Reviewer 2
We would like to thanks the positive opinion of the Reviewer about our manuscript. We accepted all remarks and suggestions. We have made changes in correction mode.
- We put some extra Reference in the manuscript as the Reviewer asked. With referenced worked we completed the motivation and the discussion. We have placed subject and the discussion in catalytic related environment. New references are marked by red in Reference List.
- Some obvious statement are clarified in the discussion.
- The funding information is corrected now.
- Spelling errors are corrected.
- The quality of figures are improved.
We very hope that the revised version is suitable for publication.

Round 2
Reviewer 2 Report
The Authors have improved the work addressing all my remarks.